# Interleukin-8 as a candidate for thymoma identification and recurrence surveillance

Shilin Gao[1,2,11], Jiahao Jiang[3,11], Chun Jin[3,4], Jian Gao[3], Dian Xiong[4], Pengjie Yang[1,5], Shuzhong Cui[5], Wenhao Yang[5], Qibin Leng [5], Jihong Dong[6], Gang Chen[7], Junzhen Liu[8], Li Wang[9], Aiwu Ke[10], Haikun Wang [1,2✉] & Jianyong Ding [3✉]

Thymoma is the most common tumor of the anterior mediastinum. Routine imaging methods such as computed tomography or magnetic resonance imaging often lead to misdiagnosis between thymoma and other thymic abnormalities. Therefore, urgently needed is to develop a new diagnostic strategy. Here we identify interleukin-8 (IL-8) as a biomarker for auxiliary diagnosis of thymoma. We find that IL-8 levels in naïve T cells are markedly elevated in patients with thymoma compared to those with other thymic tumors. IL-8 levels in naive T cells are significantly decreased after surgical resection in thymoma patients, and rise again when thymoma recurs. A receiver operating characteristic curve analysis shows that IL-8 evaluation performs well in thymoma identification, with high specificities and sensitivities. We also observe significant clinical relevance between IL-8 levels in naïve T cells and clinicopathological features. In conclusion, our study suggests that IL-8 is a biomarker for thymoma identification and recurrence surveillance.

[1] CAS Key Laboratory of Molecular Virology and Immunology, Institut Pasteur of Shanghai, Chinese Academy of Sciences, Shanghai, China. [2] University of Chinese Academy of Sciences, Beijing, China. [3] Department of Thoracic Surgery, Zhongshan Hospital, Fudan University, Shanghai, China. [4] Department of Thoracic Surgery, Xuhui Central Hospital, Shanghai, China. [5] Affiliated Cancer Hospital and Institute of Guangzhou Medical University, Guangzhou, China. [6] Department of Neurology, Zhongshan Hospital, Fudan University, Shanghai, China. [7] Department of Pathology, Zhongshan Hospital, Fudan University, Shanghai, China. [8] Department of Radiology, Zhongshan Hospital, Fudan University, Shanghai, China. [9] Physical Examination Center, Zhongshan Hospital, Fudan University, Shanghai, China. [10] Liver Cancer Institute, Key Laboratory of Carcinogenesis and Cancer Invasion, Ministry of Education, Zhongshan Hospital, Fudan University, Shanghai, China. [11] These authors contributed equally: Shilin Gao, Jiahao Jiang. ✉email: hkwang@ips.ac.cn; ding.jianyong@zs-hospital.sh.cn

Thymic masses are abnormalities in the anterior mediastinum, mainly including thymomas, thymic cysts, thymic hyperplasia, thymic carcinomas, lymphomas, and teratomas et al. Many thymic masses, such as thymomas, are thought to be malignant and should be resected if they are potentially resectable[1,2]. However, some thymic masses, such as thymic cysts, lymphomas and thymic hyperplasia in the absence of myasthenia gravis (MG) do not need surgery. Traditional imaging methods, such as computed tomography (CT) and magnetic resonance imaging (MRI), are of great importance but lacking of enough specificity on assessment of thymic masses[3,4]. The fact is that nontherapeutic thymectomy rates range from 22 to 68%[5], which is not only unnecessarily morbid to patients but also costly. Therefore, there is an urgent need for a new auxiliary diagnostic means for accurately identifying different thymic masses.

Thymus is the central lymphoid organ responsible for T-cell development, selection, and emigration[6]. In healthy adults, thymic function and thymic output gradually attenuate with aging after puberty[7,8], and therefore, newly arising T cells, also called recent thymic emigrants (RTEs), are diminished correspondingly in the peripheral blood of healthy adults. A fascinating aspect of thymic masses is that many thymic masses such as lymphocyte-rich thymomas or thymic hyperplasia have intratumorous thymopoiesis and can restore the output of newly arising T cells to some extent. Such a difference in intratumorous thymopoiesis and T-cell output may provide clues for developing new auxiliary diagnostic means.

Therefore, discovering the relationship between RTEs and pathological thymic masses is of importance. Among several markers in measuring RTEs[9–14], cytokine interleukin-8 (IL-8) performed well in clinical settings[7]. IL-8, a prototypic cytokine of innate immune cells, is produced by macrophages, some epithelial cells and endothelial cells[11]. Recent studies have revealed that IL-8 is also produced by RTEs that have most recently completed intrathymic development and egress, making it an indicator of thymic function and thymic output[7,15–17].

In this study, we analyze the proportions of IL-8 in CD4[+] and CD8[+] naïve T cells in patients with different types of thymic tumors and healthy controls in a discovery set, and validate them in an independent set with 186 cases, to evaluate the utility of IL-8 in the diagnosis and surveillance of thymoma. We find that IL-8 levels in naïve T cells are significantly elevated in patients with thymoma compared to those with other thymic tumors. IL-8 levels in naive T cells decrease after tumor removal in thymoma patients, but rise again when thymoma recurs. Our study indicates that IL-8 is a candidate biomarker for thymoma identification and recurrence surveillance.

## Results

### Discovery study of IL-8 as a candidate biomarker for thymoma.
Thymic abnormalities often affect T-cell homeostasis and function, and have been thought to be a main cause of autoimmune diseases such as MG in patients[18]. To identify candidate biomarkers for the differential diagnosis of thymic tumors, we analyzed the cytokine profiles and surface molecules in different T-cell subsets to address functional alterations in T cells from patients with different thymic tumors in a discovery set (Table 1). The diagnosis of thymic tumors was histologically confirmed in all patients involved in the discovery set. We found that the levels of IL-8 in naïve T cells were markedly elevated in thymoma group compared to any group with thymic cysts, teratomas, lymphomas or age-matched healthy controls (Fig. 1a, b, Supplementary Fig. 1). IL-8 is a signature effector cytokine of newly arising naïve T cells, also called recent thymic emigrants (RTEs)[11,13,19]. In addition to IL-8, CD31 and T-cell receptor excision circles (TRECs) are also signature markers of RTE cells[9,12,14,17]. Consistently, we observed more CD31[+] naïve CD4[+]

**Table 1 Patient characteristics.**

| Factor | Discovery set (N = 40) | Percentage | Validation set (N = 186) | Percentage |
|---|---|---|---|---|
| Age, years | | | | |
| Mean | 44.97 | | 49.46 | |
| Range | 20–77 | | 20–77 | |
| Sex | | | | |
| Male | 20 | 50% | 88 | 47.2% |
| Female | 20 | 50% | 98 | 52.7% |
| Healthy controls | 10 | 25% | 33 | 17.8% |
| Thymic mass | | | | |
| Thymoma | 10 | 25% | 68 | 36.6% |
| Thymic cyst | 5 | 12.5% | 48 | 25.8% |
| Thymic hyperplasia | 0 | 0% | 2 | 1.1% |
| Thymic carcinoma | 5 | 12.5% | 13 | 7.0% |
| Teratoma | 5 | 12.5% | 14 | 7.5% |
| Lymphoma | 5 | 12.5% | 8 | 4.3% |
| Thymoma WHO subtype | | | | |
| A | 0 | 0% | 2 | 2.9% |
| AB | 3 | 30% | 19 | 27.9% |
| B1 | 0 | 0% | 3 | 4.4% |
| B2 | 4 | 40% | 26 | 38.2% |
| B3 | 3 | 30% | 18 | 26.4% |
| Thymoma Masaoka stage | | | | |
| I | 1 | 10% | 10 | 14.7% |
| II | 5 | 50% | 23 | 33.8% |
| III | 3 | 30% | 25 | 36.8% |
| IVa | 1 | 10% | 10 | 14.7% |
| Myasthenia gravis in thymoma patients | | | | |
| Yes | 2 | 20% | 26 | 38.2% |
| No | 8 | 80% | 42 | 61.8% |

T cells in thymoma patients compared to patients with other thymic tumors including thymic cysts, thymic carcinomas, teratomas, and lymphomas (Supplementary Fig. 2a, b), and elevated sjTRECs levels in naïve T cells in thymoma patients compared to age-matched healthy controls or patients with patients with other thymic tumors (Supplementary Fig. 2c), indicating more RTE cells in thymoma patients.

RTE cells have been reported to be tightly associated with thymopoiesis in humans, with a dramatic decrease in RTE cells after thymus resection[7]. Therefore, we next detected intratumorous thymopoiesis in thymic tumors surgically removed from patients in the discovery set. It was very consistent that patients with thymomas, but not patients with thymic cysts, thymic carcinomas, teratomas, or lymphomas, had potent intratumorous thymopoiesis (Supplementary Fig. 3). Our data suggest that the proportion of IL-8[+] naïve T cells reflects intratumorous thymopoiesis and naïve T-cell outputs from thymic tumors, and thus IL-8 may emerge as a biomarker candidate for differential diagnosis of thymic tumors.

### The utility of IL-8 as a biomarker of thymoma.
We then evaluated IL-8 as a biomarker for the differential diagnosis of thymic masses in an independent validation set of 186 cases that were not included in the discovery set. The validation set included healthy controls, and five different patient groups according to the pathological diagnosis: thymomas, thymic cysts, thymic carcinomas, teratomas, lymphomas, and thymic hyperplasia (Table 1). Blinded analysis confirmed that both the proportion of IL-8[+] naïve CD4[+] T cells and the proportion of IL-8[+] naïve CD8[+] T cells were markedly elevated in patients with thymomas, compared to healthy controls or patients with thymic cysts, teratomas, and lymphomas in the validation set (Fig. 1c, d). In-line with findings in the discovery set, CD31[+] naïve T cells were increased in patients with thymomas (Supplementary Fig. 4a), reaffirming more RTE cells in thymoma patients.

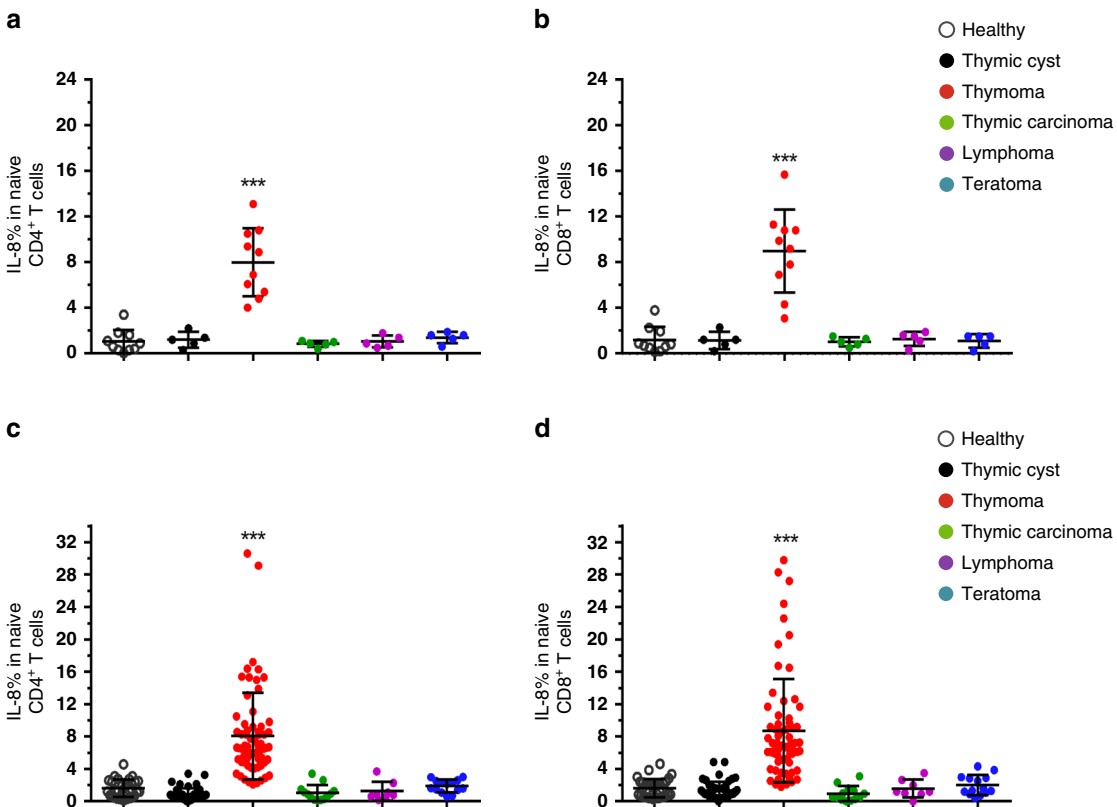

**Fig. 1 The proportions of IL-8⁺ naïve T cells are increased in thymoma patients.** IL-8 and CD31 expression in naïve T cells (CD15⁻CD14⁻CD3⁺TCRαβ⁺ CD45RA⁺CCR7⁺) in peripheral blood mononuclear cells (PBMCs) from patients with different thymic tumors (thymomas, thymic cysts, thymic carcinomas, lymphomas, and teratomas) and age-matched healthy controls were analyzed by Flow cytometry. **a, b** The proportions of IL-8⁺CD31⁺ cells in **a** CD4⁺ naïve T and in **b** CD8⁺ naïve T cells in the discovery set (healthy controls, $n = 10$; thymic cysts, $n = 5$; thymomas, $n = 10$; thymic carcinomas, $n = 5$; lymphomas, $n = 5$; teratomas, $n = 5$). **c, d** The proportions of IL-8⁺CD31⁺ cells in **c** CD4⁺ naïve T cells and in **d** CD8⁺ naïve T cells in the validation set (healthy controls, $n = 33$; thymic cysts, $n = 48$; thymomas, $n = 68$; thymic carcinomas, $n = 13$; lymphomas, $n = 8$; teratomas, $n = 14$). Data in **a–d** are shown as the mean ± standard deviation (SD). Statistical differences were determined by two-sided Krusal–Wallis Analysis of Variance test and adjusted with the Benjamini–Hochberg procedure. Adjusted $P$ values were indicated by *** (adjusted $p < 0.001$). Source data are provided as a Source Data file.

We followed upon IL-8 levels of naïve T cells in patients after surgery. The proportions of IL-8⁺ naïve T cells were dramatically reduced in thymoma patients 3 months after tumor removal, whereas patients with thymic cysts, thymic carcinomas, teratomas, and lymphomas showed no obvious alteration in the proportion of IL-8⁺ naïve T cells after surgical resection (Fig. 2), suggesting that the increase in the proportion of IL-8⁺ naïve T cells in thymoma patients is caused by thymoma.

We also observed high levels of IL-8 in blood naïve T cells in two thymic hyperplasia patients with myasthenia gravis. The levels of IL-8 were subsequently decreased after surgical resection for the treatment of myasthenia gravis (Supplementary Fig. 5), suggesting that thymic hyperplasia can also result in the elevated IL-8 levels in naïve T cells. This observation supports our notion that IL-8 levels in naïve T cells are tightly associated with the extent of T-cell outputs from thymic masses.

Given the differential proportions of IL-8⁺ naïve T cells from patients with different thymic tumors, we next asked whether evaluation of the proportion of IL-8⁺ naïve T cells could be a good auxiliary method for the differential diagnosis between thymomas and other thymic tumors. Therefore, we performed a receiver operating characteristics (ROC) curve analysis to evaluate the diagnostic sensitivity and specificity of IL-8 as a biomarker of thymomas. According to the ROC curve analysis, the area under the ROC curve for the proportion of IL-8⁺ naïve CD4⁺ T cells versus a diagnosis of thymoma was 0.98, with a 95%

confidence interval (CI) ranging from 0.95 to 0.99 (Fig. 3a); such a high area under the curve (AUC) was also yielded for IL-8⁺ naïve CD8⁺ T cells in the diagnosis of thymoma (Fig. 3b). As determined by Youden's index, the optimal cutoff point of the proportion of IL-8⁺ naïve CD4⁺ T cells for the diagnosis of thymoma was 2.66% (95% CI for cutoff point: 2.225–3.945%), with 94.1% specificity and 92.8% sensitivity (Fig. 3c); the optimal cutoff point of the proportion of IL-8⁺ naïve CD8⁺ T cells for the diagnosis of thymoma was 3.35% (95% CI for cutoff point: 1.775–5%), with 94% specificity and 88.2% sensitivity (Fig. 3d). In addition to IL-8 assay, we also evaluated the diagnostic performances with other RTE markers including CD31, PTK7, and CR2 in some patients. Compared to IL-8 assay, CD31 detection had much lower diagnostic sensitivity and specificity in distinguishing thymomas from other types of thymic tumors (Supplementary Fig. 4b, c); PTK7 or CR2 assay also showed lower discriminatory ability in thymoma diagnosis, only with AUC level 85% and 84%, respectively (Supplementary Fig. 6 and 7). Taken together, our data revealed that IL-8 in naïve T cells is a reliable diagnostic biomarker to distinguish patients with thymomas from those with other types of thymic tumors. Of note, IL-8 evaluation alone might not distinguish thymoma from thymic hyperplasia, which remains to be studied. However, given that thymic hyperplasia has some typical imaging manifestations on CTs/ MRIs[20], IL-8 evaluation combined with chest CTs/MRIs can accurately identify thymoma and thymic hyperplasia.

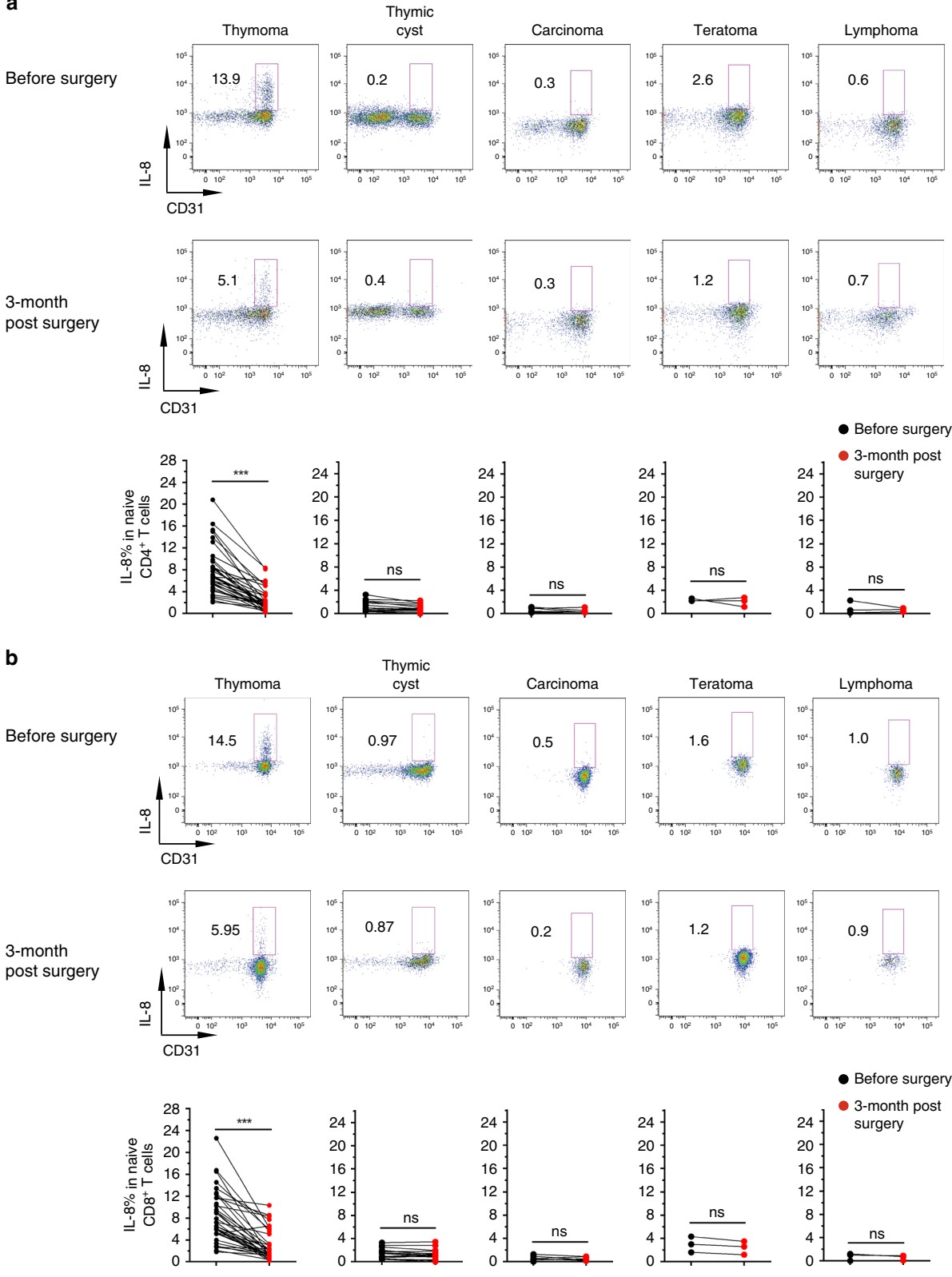

**IL-8 in the surveillance of thymoma recurrence**. In the follow-up process, we observed tumor recurrence in four thymoma patients by CT inspection. In these patients, tumor removal in the first operation led to an obvious decrease in the proportions of IL-8[+] naïve T cells 3 months after surgery; however, the proportions of IL-8[+] naïve T cells increased again with tumor recurrence (Fig. 4, Supplementary Fig. 8). We also observed gradually decreased proportions of IL-8[+] naïve T cells during follow-up after the second operation (Fig. 4a, b, Supplementary Fig. 8a, b). These four special cases suggested that IL-8 could be applied to the active surveillance of thymoma recurrence after surgical resection.

**Fig. 2 The increase in IL-8⁺ naïve T cells in thymoma patients is caused by thymic tumors.** The proportions of IL-8⁺ naïve T cells in patients with different thymic tumors (thymomas, thymic cysts, thymic carcinomas, lymphomas, and teratomas) were measured before and 3 months post thymic surgeries. **a** Representative flow-cytometry plots of IL-8⁺CD31⁺ cells in CD4⁺ naïve T cells in PBMCs from patients before and after tumor removal (upper panel); summary of the frequencies of IL-8⁺CD31⁺ in CD4⁺ naïve T cells in PBMCs from patients before and after tumor removal (lower panel) (thymomas, $n = 35$; thymic cysts, $n = 18$; thymic carcinomas, $n = 6$; lymphomas, $n = 3$; teratomas, $n = 3$). **b** Representative flow-cytometry plots of IL-8⁺CD31⁺ cells in CD8⁺ naïve T cells in PBMCs from patients before and after tumor removal (upper panel); summary of the frequencies of IL-8⁺CD31⁺ CD8⁺ naïve T cells in PBMCs from patients before and after tumor removal (lower panel) (thymomas, $n = 35$; thymic cysts, $n = 18$; thymic carcinomas, $n = 6$; lymphomas, $n = 3$; teratomas, $n = 3$). Numbers adjacent to the outlined areas of flow-cytometry plots in **a, b** indicate the percentages of IL-8⁺ naïve T cells. Dot plots, each point represents an individual patient. Statistical differences were determined by Wilcoxon test and adjusted with the Benjamini–Hochberg procedure. Adjusted P values were indicated by *** (adjusted $p < 0.001$), or n.s. (adjusted $p > 0.05$). Source data are provided as a Source Data file.

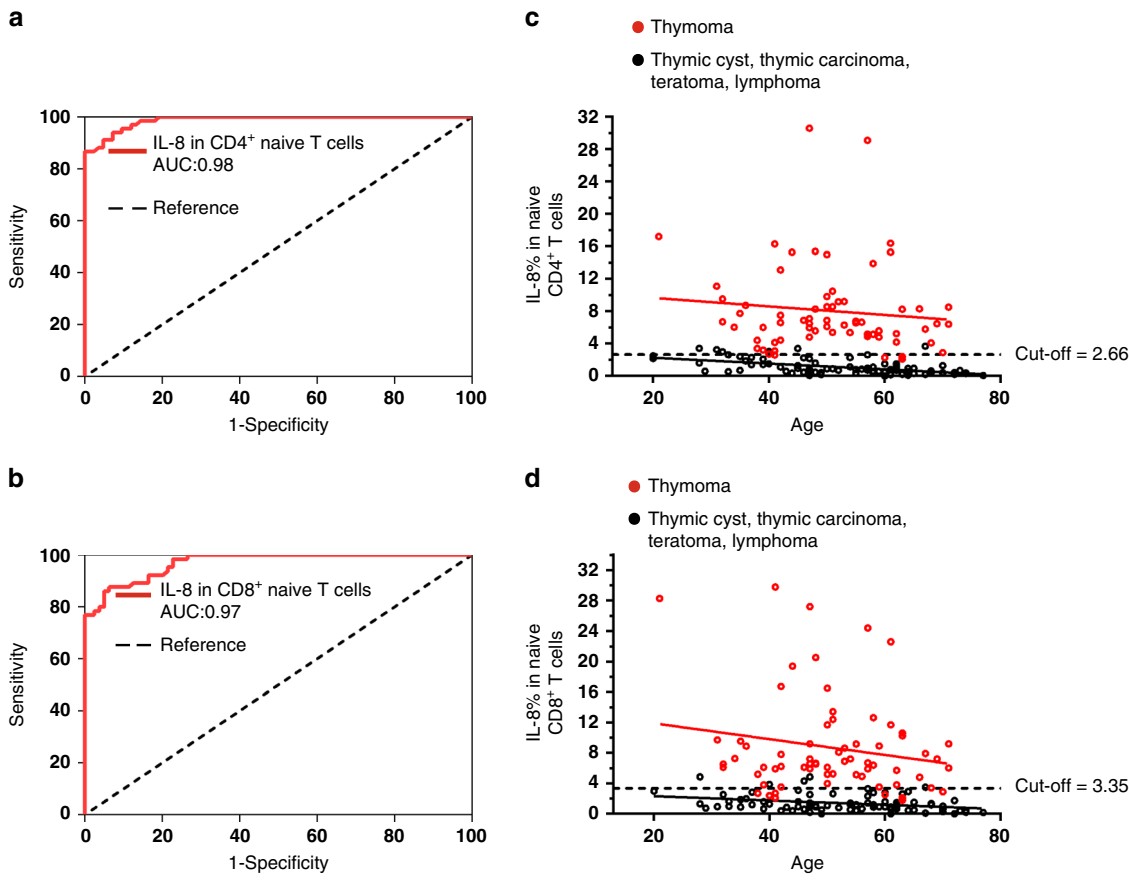

**Fig. 3 IL-8 is a reliable diagnostic biomarker to distinguish thymomas from other types of thymic tumors.** Receiver operating characteristics (ROC) curve analyses were performed to evaluate the diagnostic sensitivity and specificity of IL-8 as a biomarker of thymomas. **a** Diagnostic performance of IL-8 in CD4⁺ naïve T cells in identifying patients with thymomas ($n = 68$) from patients with other thymic tumors ($n = 83$) assessed by ROC curve analysis. **b** Diagnostic performance of IL-8 in CD8⁺ naïve T cells in identifying patients with thymomas ($n = 68$) from patients with other thymic tumors ($n = 83$) assessed by ROC curve analysis. **c** The frequencies of IL-8⁺CD4⁺ naïve T cells in patients with thymomas ($n = 68$) and other thymic tumors ($n = 83$) of various ages. The dashed line indicates the optimum cutoff point calculated by applying Youden's J statistic to ROC curves. **d** The frequencies of IL-8⁺ CD8⁺ naïve T cells in patients with thymomas ($n = 68$) and other thymic tumors ($n = 83$) of various ages. The dashed line indicates the optimum cutoff point calculated by applying Youden's J statistic to ROC curves. Source data are provided as a Source Data file.

**Clinical relevance of IL-8 in thymomas.** Next, we asked whether there is clinical relevance between the proportion of IL-8⁺ naïve T cells and clinicopathological features, including WHO subtypes, Masaoka stages, the presence of MG and tumor sizes, in patients with thymomas. Statistical analyses revealed that the frequencies of IL-8-positive T cells in patients with "lymphocyte-rich" type B2 thymomas are higher than those in patients with "lymphocyte-poor" type A and B3, confirming that IL-8 positive RTE cells are tightly associated with the potency of intratumorous thymopoiesis (Table 2, Supplementary Fig. 9). The proportions of IL-8⁺ naïve

T cells were higher in thymoma patients at Masaoka stage III and IV than in patients at stage I and II (Table 2, Supplementary Fig. 9). Elevated IL-8 levels in naïve T cells in patients at Masaoka stage III and IVa might be explained by the fact that the majority of Masaoka stage III and IVa thymomas are type B2 thymomas[21]. We also found that tumor size was positively correlated with the proportion of IL-8⁺ naïve T cells, as patients with larger thymomas had higher proportion of IL-8⁺ naïve T cells (Table 2, Supplementary Fig. 9). Myasthenia gravis is an important paraneoplastic manifestation of patients with thymomas. In our study, we did not find any

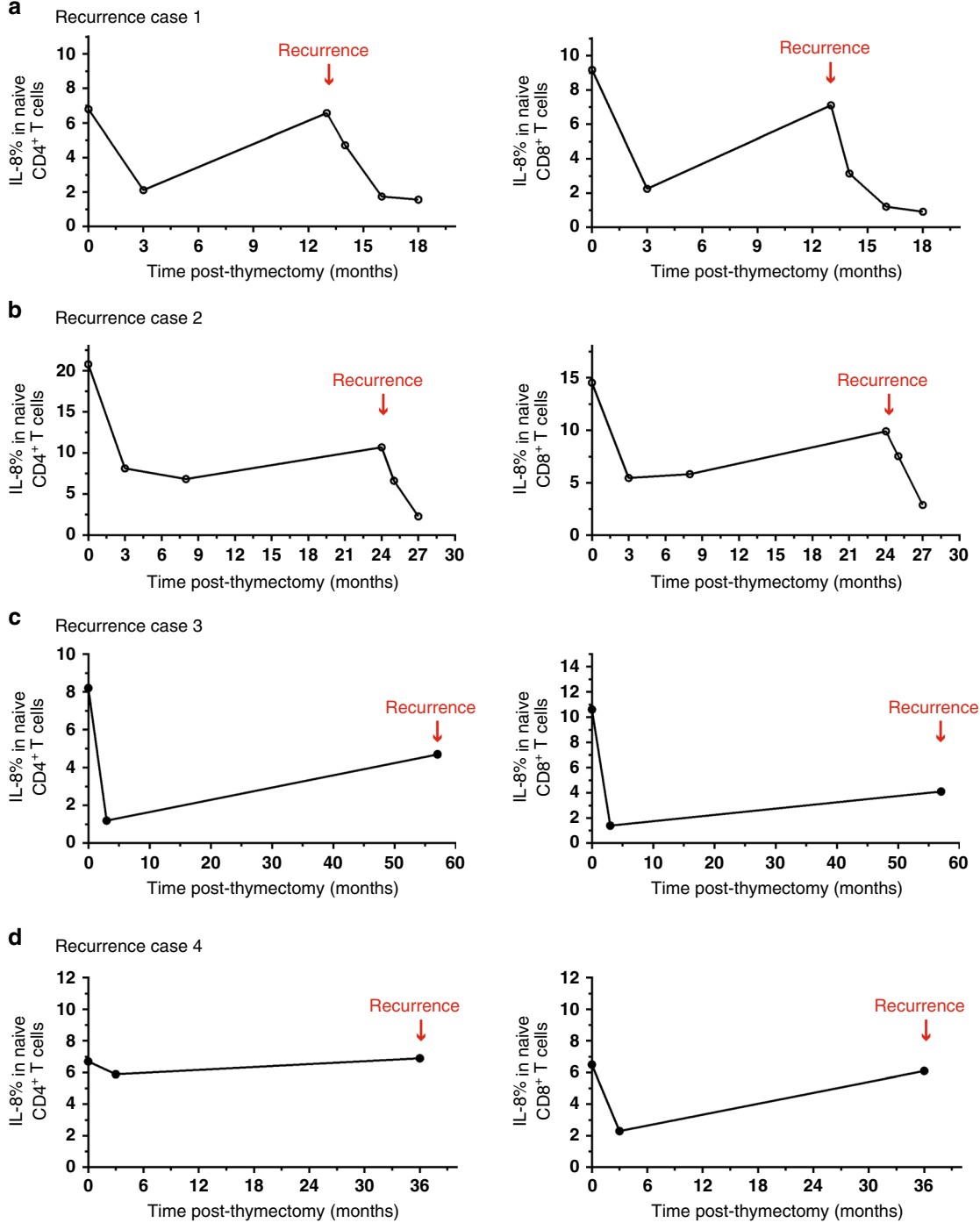

**Fig. 4 The utility of IL-8 in the surveillance of thymoma recurrence. a** The dynamic proportions of IL-8+CD31+ cells in CD4+ naïve T cells (left panel) or CD8+ naïve T cells (right panel) in PBMCs at different time points before and after thymoma resection in Case #1 patient with thymoma recurrence. **b** The dynamic proportions of IL-8+CD31+ cells in CD4+ naïve T cells (left panel) or CD8+ naïve T cells (right panel) in PBMCs at different time points before and after thymoma resection in Case #2 patient with thymoma recurrence. **c** The dynamic proportions of IL-8+CD31+ cells in CD4+ naïve T cells (left panel) or CD8+ naïve T cells (right panel) in PBMCs at different time points before and after thymoma resection in Case #3 patient with thymoma recurrence. **d** The dynamic proportions of IL-8+CD31+ cells in CD4+ naïve T cells (left panel) or CD8+ naïve T cells (right panel) in PBMCs at different time points before and after thymoma resection in Case #4 patient with thymoma recurrence. Source data are provided as a Source Data file.

obvious difference in the proportion of IL-8+ naïve T cells between thymoma patients with and without MG. In addition, IL-8 was not associated with sex or age.

Collectively, our data suggest that there is significant clinical relevance between IL-8 levels in naïve T cell and clinicopathological features.

## Discussion

This study casts an insight into the initial evaluation of IL-8 as a biomarker for patients with thymoma. By traditional imaging methods, many thymic masses, such as thymic cysts and lymphomas, are often misdiagnosed as thymoma, which leads to unnecessary surgical resection. Our data have proven that thymomas can

**Table 2 Clinical relevance of IL-8⁺ naïve T cells in patients with thymomas.**

| Factor | IL-8 in CD4⁺ naïve T cells | Adjusted P value[a] | IL-8 in CD8⁺ naïve T cells | Adjusted P value[a] |
|---|---|---|---|---|
| Sex | | | | |
| Male | 9.9 ± 6.9 | 0.2014 | 10.2 ± 6.8 | 0.4573 |
| Female | 7.6 ± 4.4 | | 8.7 ± 6.7 | |
| WHO pathological type | | | | |
| A, AB and B1 | 6.3 ± 3.1 | 0.0444 | 6.3 ± 3.8 | 0.0253 |
| B2, B3 | 8.9 ± 6.0 | | 10.0 ± 7.0 | |
| Myasthenia gravis | | | | |
| Yes | 8.5 ± 5.8 | 0.5638 | 8.7 ± 6.1 | 0.9558 |
| No | 7.4 ± 4.0 | | 8.8 ± 6.8 | |
| Masaoka stage | | | | |
| I–II | 6.4 ± 3.2 | 0.0249 | 6.7 ± 4.1 | 0.0249 |
| III–IVa | 9.5 ± 6.6 | | 10.6 ± 7.6 | |
| Age, years | | | | |
| 20–40 | 9.9 ± 5.6 | 0.4270 | 10.7 ± 7.8 | 0.5803 |
| 40–50 | 9.4 ± 6.4 | | 10.5 ± 8.0 | |
| 50–60 | 8.2 ± 5.8 | | 8.7 ± 4.8 | |
| >60 | 6.9 ± 4.0 | | 7.3 ± 5.1 | |
| Tumor size, cm | | | | |
| 2–4 | 6.2 ± 1.3 | <0.0001 | 6.0 ± 2.1 | <0.0001 |
| 4–6 | 6.6 ± 2.8 | | 6.3 ± 2.6 | |
| 6–8 | 9.2 ± 3.9 | | 8.6 ± 2.8 | |
| 8–10 | 12.0 ± 8.6 | | 13.4 ± 7.8 | |
| >10 | 16.3 ± 6.5 | | 19.2 ± 6.8 | |

[a]P values were adjusted with the Benjamini–Hochberg procedure.

be accurately distinguished from other thymic masses by IL-8 evaluation. Thus, the application of IL-8 evaluation could reduce the rate of unnecessary or nontherapeutic surgical resection of thymic masses. It is known that different thymic masses have some typical imaging manifestations on CTs/MRIs[20], therefore, IL-8 evaluation combined with CT/MRI could identify the vast majority of thymic masses.

Thymopoiesis is active in children but considerably reduced in adults due to thymic involution[22,23]. However, adult patients with thymomas and thymic hyperplasia possess the capacity of thymopoiesis in tumor or hyperplastic thymic tissues, and generate and export more naïve T cells[24,25]. Accumulating studies have shown that IL-8 is a signature cytokine of newly arising T cells from the thymus in healthy humans. In our study, we showed that the IL-8 levels in naïve T cells are elevated in patients with thymomas, suggesting that naïve T cells recently generated from thymomas also express high levels of IL-8. Consistent with these findings, the surgical removal of thymomas leads to a gradual decrease in the proportion of IL-8⁺ naïve T cells.

In addition to IL-8, other RTE markers might be applied to differentiate thymomas from the variety of thymic tumors. In our study, the RTE marker CD31 showed lower diagnostic sensitivity and specificity than IL-8, which could be explained by the fact that not all CD31⁺ naïve T cells are RTE cells[26]. As the "golden standard" of RTEs, the levels of sjTRECs do increase in patients with thymoma in our study and studies by others[27,28]. Unfortunately, sjTRECs quantification is difficult to apply in clinical settings[26,29], due to high cost and complex performance. Recently, several new human surface RTE markers, such as PTK7 and CR2 have been identified[9,13]. Although analysis of surface markers is more direct compared to intracellular staining of IL-8 in T cells, these markers show lower diagnostic accuracy in thymoma diagnosis in our study, when compared with IL-8 assay. Therefore, so far IL-8 is a relatively more specific biomarker for thymoma diagnosis.

The presence of myasthenia gravis (MG) is a unique feature of patients with thymoma[21,30]. Actually, a large number of patients are diagnosed with thymoma based on the appearance of MG symptoms, such as eyelid drooping and dysphagia. In addition to

thymoma, thymic hyperplasia can also cause MG[31]. In contrast, other thymic tumors, including thymic cysts, thymic carcinoma, teratoma, and lymphoma, hardly cause MG. Although the cause of MG is still unclear, we suspect that it may be related with thymopoiesis state, because surgical resection can improve the clinical outcomes of patients with nonthymomatous MG[32]. We noticed that previous report declared that MG is highly associated with the efficiency of thymomas to produce and export naive CD4⁺ T cells[18]. Other report, on the contrary, stated that thymomas with MG exhibit higher level of naïve CD8⁺ T cells rather than naïve CD4⁺ T cells[33]. However, we did not find any obvious difference in the proportion of IL-8⁺ naïve T cells between thymoma patients with and without MG. This phenomenon may be attributed to immunosuppressive drugs that are used prior to thymectomy. Given that IL-8 levels in naïve T cells can reflect the thymopoiesis state, the elevated IL-8 level in naïve T cells could be as a useful factor to determining whether patients with MG need a thymectomy, especially for the patients with not typical thymoma or small thymoma. Further study is needed to address this issue.

In our study, we found that the proportion of IL-8⁺ naïve T cells in patients is associated with many factors, such as tumor size, histological classification, and Masaoka clinical stage. Thymomas with larger tumor sizes and later clinicopathologic stages have a higher proportion of IL-8⁺ naïve T cells. We attribute this phenomenon to the scale (tumor size) and potency (WHO classification) of intratumorous thymopoiesis. On the other hand, the proportion of IL-8+ naïve T cells is also associated with the Masaoka stage, high proportion of IL-8⁺ naïve T cells in patients with Masaoka stage III and IVa thymomas. This result might be explained by the fact that the majority of Masaoka stage III and IVa thymomas are type B2 thymomas (according to a previous studies[21] and our data).

Currently, active surveillance with serial CT or MRI is commonly used to detect thymoma recurrence after surgical resection. In our study, we observed tumor recurrence in four thymoma patients. The proportion of IL-8⁺ naïve T cells were dramatically reduced after surgical resection but increased again when the thymoma

recurred. Therefore, IL-8 could be a useful marker to monitor postoperative tumor recurrence in patients with thymomas.

Our study still has some limitations. Thymomas typically occur in adults from age 40 to 70, but a few thymomas can be found in patients under 40 years old[1,3]. Our study shows more borderline cases in IL-8 assay in patients aged 40 years and younger, compared to patients aged over 40 years. Therefore, IL-8 and other RTE markers evaluation may not work well in not only children and adolescents but also adults with remaining thymic tissues (up to 40 years old). In addition, IL-8 evaluation alone seems not to work for differential diagnosis between thymoma and thymic hyperplasia. IL-8 evaluation should be combined with chest CTs/MRIs for accurate identification between thymoma and thymic hyperplasia.

Based on the above findings and National Comprehensive Cancer Network (NCCN) guidelines, we present a proposal for the diagnostic procedure for thymic masses (Supplementary Fig. 10). Although chest CTs/MRIs with contrast play an important role in screening incidentally detected thymic masses and evaluating tumor staging, it is difficult for CTs/MRIs to discriminate thymomas from thymic cysts and lymphomas. The proportion of IL-8[+] naïve T cells can discriminate them accurately, and thus, IL-8 evaluation could be a reliable auxiliary method for the differential diagnosis of thymic masses[5]. Although IL-8 evaluation alone might not be used for differential diagnosis between thymoma and thymic hyperplasia, chest CTs/MRIs with contrast can easily differentiate them. Thus, chest CTs/MRIs with contrast combined with the proportion of IL-8[+] naïve T cells from the peripheral blood could identify most thymic masses.

In summary, our findings demonstrate that IL-8 has significant diagnostic value as a biomarker for identifying thymomas and monitoring thymoma recurrence.

## Methods

**Study design.** This discovery-based study was designed to identify IL-8 as a biomarker for the differential diagnosis of thymic masses to complement traditional imaging methods. To determine the clinical relevance between IL-8 and different thymic masses, this study analyzed IL-8 levels in naïve T cells in peripheral blood mononuclear cells (PBMCs) from patients with thymic masses before and after surgical resection. All patients enrolled in this study underwent surgical resection for thymic masses, including thymomas, thymic cysts, thymic hyperplasia, thymic carcinomas, teratomas, and lymphomas, between December 2015 and May 2020 in the Thoracic Surgery Department, Zhongshan Hospital, Fudan University (Shanghai, China) and Xuhui Central Hospital (Shanghai, China). Heparinized blood samples were collected before surgery. Tissues, including tumor tissues and paratumor tissues, were collected during surgery. Healthy controls were collected from the Physical Examination Center, Zhongshan Hospital (Shanghai, China). To avoid overfitting, we used flow cytometry to screen biomarker candidates of thymic masses in a discovery set of 40 samples (30 patients and 10 healthy controls) and validated them in an independent set of 186 samples (153 patients and 33 healthy controls). To reduce bias, all samples were handled, stored and processed under the same procedure. Written informed consent was obtained from each patient, and ethical approval was obtained from the Zhongshan Hospital Research Ethics Committee and Xuhui Central Hospital Research Ethics Committee.

**Discovery set and validation set.** Patients with thymomas, thymic cysts, thymic carcinomas, teratomas, and lymphomas, and healthy controls were included in the discovery set. All cases in the discovery set were reviewed by senior pathologists.

Patients in the validation set were independent from those in the discovery set. The validation set consisted of patients with thymomas, thymic cysts, thymic hyperplasia, thymic carcinomas, teratomas, and lymphomas, and healthy controls. The ages and donor sources of patients in the validation set were the same as those of patients in the discovery set. All cases in the validation set were reviewed by senior pathologists. Patient information is shown in Table 1.

**Isolation of PBMCs and thymocytes.** PBMCs were isolated from heparinized blood samples using Lymphoprep (Stem Cell Technologies #07861) density gradient centrifugation at $500 \times g$ for 30 min at room temperature. PBMCs were then washed twice with T-cell medium (Dulbecco's Modified Eagle's Medium (DMEM) containing 10% heat-inactivated fetal bovine serum (FBS), 2 mM L-glutamine, penicillin–streptomycin, nonessential amino acids, sodium pyruvate, vitamins, 10 mM HEPES, and 50 μM 2-mercaptoethanol), and resuspended in T-cell medium.

For thymocytes isolation, thymus specimens including thymic tumor tissues and paratumor tissues were kept on ice after surgical removal. Visible blood vessels were removed and tissues were washed with ice-cold DMEM medium. Tissues were minced by sterilized scissors, and meshed in a sterile 70-μm cell strainer. Then, the strainer was rinsed with 5 ml cold DMEM medium, and the single-cell suspension was filtered again using a 70-μm cell strainer to remove large debris. Cells were washed twice with cold DMEM medium, and resuspended with T-cell medium.

**Flow-cytometry analysis.** For surface-marker staining, $1 \times 10^6$ PBMCs were washed with staining buffer (phosphate-buffered saline (PBS) with 1% FBS, and blocked with Fc-blocker and 10% rat serum for 5 min on ice. Cells were then stained with surface-marker antibodies (listed in the Supplementary Table 1) for 30 min on ice, followed by wash with staining buffer. Samples were analyzed with a FACSFortessa instrument (BD).

For IL-8 staining, PBMCs ($2 \times 10^6$) were stimulated with phorbol myristate acetate (PMA, 10 ng ml$^{-1}$, Sigma) and ionomycin (0.5 μg ml$^{-1}$, Sigma) for 6 h at 37 °C in T-cell medium containing 10% FBS, and 5 μg ml$^{-1}$ Brefeldin A (BFA, Sigma) was added for the last 2 h of stimulation. Stimulated cells were collected and blocked with Fc-blocker and 10% rat serum for 5 min on ice, followed by staining with surface-marker antibodies (listed in the Supplementary Table 1) for 30 min on ice. Cells were then washed with PBS, fixed with 3.7% paraformaldehyde (15 min, RT), and permeabilized with 0.2% saponin (15 min on ice). Cells were stained with anti-IL-8 antibody on ice for 30 min, and washed twice with 0.2% saponin buffer. Finally, samples were analyzed with a FACSFortessa instrument (BD). Gating strategy is shown in Supplementary Fig. 11.

**sjTREC quantification.** DNA was extracted from sorted naïve CD4$^+$ T cells and naïve CD8$^+$ T cells with QIAamp DNA Blood Mini Kit (Qiangen) according to manufacturer's protocol. SjTRECs were analyzed by the real-time PCR method as previously described[34]. Briefly, each PCR reaction contained 10 μl Premix Ex Taq (Takara), 0.4 μl ROX Reference Dye, 0.8 μl oligonucleotide primers (Supplementary Table 2), 0.8 μl FAM-TEMRA labeled probe (Supplementary Table 2), 6 μl dH₂O and 2 μl DNA sample. As internal standards, TREC (T-cell receptor excision circles) and TCRAC (T-cell receptor alpha constant) DNA sequences from human thymocytes were cloned into vector pcDNA3.0, respectively[34], and quantified with samples simultaneously. The copies of sjTRECs per $10^6$ naïve CD4$^+$ T cells and naïve CD8$^+$ T cells were calculated by setting the following formula appropriately: [(mean quantity of TRECs)/(mean quantity of TCRACs/2)]×$10^6$.

**Statistical analysis.** Statistical analyses were performed and presented using Prism 6 software (GraphPad) and SPSS 19.0. The statistical significance between two groups was assessed using Mann–Whitney U test for unpaired data and Wilcoxon's signed rank test for paired data. For multiple-group comparisons, the Kruskal–Wallis Analysis of Variance test (two-sided) was used. ROC curve analyses were performed by R to determine the diagnostic capability. The 95% CI for the cutoff values were calculated based on nonparametric Bootstrap method by using R (version 3.3.3)[35,36]. Benjamini–Hochberg procedure was used to adjust $p$ values, to control the False Discovery Rate in this study. Statistical significance levels were reported as follows: n.s. for adjusted $p > 0.05$; "*" for adjusted $p < 0.05$; "**" for adjusted $p < 0.01$; "***" for adjusted $p < 0.001$ or less.

**Reporting summary.** Further information on research design is available in the Nature Research Reporting Summary linked to this article.

## Data availability

All data supporting the findings of this study are available in the article, supplementary information, or from the corresponding authors upon reasonable request. Source data underlying Figs. 1–4, Table 2 and Supplementary Figs. 2, 4–9 are provided in the Source Data file.

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

## Acknowledgements

This work was supported by the following grants: Strategic Priority Research Program of the Chinese Academy of Sciences (XDB29030103, H.W.), National Key R&D Program of China (2016YFA0502202, H.W.), and the National Natural Science Foundation of China (81972168, J.D.). We thank Xiaowei Ren and Qinghua Liao for helping us with statistical analysis.

## Author contributions

H.W. and J.D. conceived and supervised the research; S.G., J.J., C.G., D.J., L.Q., Y.W., C.Z., S.C., A.K., W.Y., Q.L., J.D., and H.W. contributed to the design of the project and discussions; S.G., J.J., G.J., Y.P., and J.C. performed the experiments; J.D., C.J., J.J., J.G., and D.X. contributed to materials or clinical data; S.G., J.J., J.D., and H.W wrote the manuscript.

## Competing interests

The authors declare no competing interests.
