## [Peer Review File · Nature Communications]

Reviewers' comments:

Reviewer #1 (Remarks to the Author):

1. Were sample size and power calculations performed for the study? If yes, please describe them in the methods section. If not, please state so in the methods section.
2. Please add percentages in the categorical variables presented in Supplementary Table 1. This will enable the reader to compare the composition of the discovery and validation sets. Please check the table for typos. For example the total number of males and females in the validation set exceeds 180.
3. The manuscript presents a large number of formal statistical comparisons and corresponding p-values without addressing the issue of multiplicity of inferences and controlling the FWER in the study. Please address this issue. For discovery studies, it would be more appropriate to consider a metric such as False Discovery Rate.
4. Please provide an estimate of the uncertainty about the cutoff value computed using Yuden's index. Almost none of the IL-8 values in the non-thymoma group reached the threshold in Fig3c.
5. The recommendation presented in Fig 5 includes the use of imaging studies. However, the manuscript does not present information about the combination of imaging with the markers evaluated here.

Reviewer #2 (Remarks to the Author):

I have read with interest the manuscript entitled: "Identification of Thymoma with the Biomarker Cytokine Interleukin-8".

I think the presented data may support author's conclusions however the manuscript would strongly benefit from far more detailed presentation of the T-cell data as well as an improved (more detailed) methodology section.

I like the simplicity of the study although find it quite predictable that some thymoma patients display increased production of IL-8 by RTEs T cells despite increased age as compared to age-matched controls (calling it a specific biomarker is another question). Indeed, if well-validated increased levels of RTEs in older patients may be a useful biomarker of specific thymic abnormalities. I speculate, however, that due to extensive variability among different thymoma patients, its usefulness may be limited (especially in some border line cases) and predominantly may apply as a progression marker when samples from the individual patient could be compared before and after thymic surgery. I stress however that I am not a thoracic clinician.

Clinical application of an assay that visualises increased frequency of RTEs as a useful biomarker for thymoma requires in-vitro activation of T cells and intracellular IL-8 staining protocol, which extends to the minimum time of at least 8 hours of processing (is it really a useful in a clinic? How variable the assay may be?).

As authors mentioned there are other markers of recent thymic emigrants and it is essential that those are also incorporated into the study, as they may pose more practical in a clinical setting. Authors discuss the need to test extra surface RTE-markers however they do not address this question (in at least limited number of additional experiments). Authors only test (and compare to IL-8) one surface marker of RTEs: CD31 (PECAM1). CD31 however, has been already demonstrated not to be very useful as a surface marker of bona fide RTEs, based on the fact that CD31+ cells already contain a population of peripherally expanded naïve T cells (based on dilution of TCR excision circles (TRECs)). Furthermore, CD31+ T cells also contain a subpopulation of

CD45RA negative memory cells. It is therefore not surprising that surface levels of CD31 do not correlate as well with marking thymomas as does the IL-8 protein expression by RTEs.

Therefore, it would be much more useful to apply direct surface staining (directly on the whole blood and not even PBMCs) using a combination of naïve T cell markers and RTE-markers previously demonstrated to identify cells expressing IL-8 (CXCL8) such as PTK7 as well as complement receptors 2 (CR2). CR2 has been demonstrated to mark RTEs (also in adults what is especially important when thymoma patients are studied), importantly, when visualized with the bright fluorochrome such as PE and the published CR2 ab clone. CR2 levels will drop after activation and therefore it cannot be used to gate cells secreting IL-8 directly. However, it (IL-8 staining) would not be needed if surface staining biomarker of RTEs would be proven useful (it would streamline the clinical stratification process).

I find it somehow surprising that followed activation of T cells (6 hours of PMA/ionomycin) authors are still able to use CD31 protein as a gating marker for RTE (part of a gating strategy) as its (PECAM1) levels decrease following activation in-vitro (at least at the RNA level).

I wish I could see a more detailed presentation of the raw data, gating strategy of activated and control RTEs: starting from all acquired events through CD3, TCRab, CD4, CD8, CD45RA and CD31. This concerns my point above (CD31 down-regulation following activation) as well as for visualisation of possible CD4/CD8 double-positive (DP) population also in the blood of thymoma patients.

There is an extensive variability within the frequency of IL-8 positive T cells among the group of thymoma patients- I wish authors could better describe what could potentially explain this variability, as currently they only discuss this matter without detailed data presentation of IL-8+ T cells frequency vs different variables studied (tumour mass etc). Paper will benefit from a more advanced description of thymoma stages and perhaps histopathological data.

Overall I think the manuscript contains interesting scientific and clinical information and has a potential for publication when all questions are carefully addressed.

Best of luck

Reviewer #3 (Remarks to the Author):

The manuscript by Gao et al. describes the use of IL8 in naïve T cells in the circulation as a way to distinguish thymomas and thymic hyperplasia from other diagnoses in the mediastinum (lymphomas, germ cell tumors and thymic cysts). Of interest the levels of IL8 in CD4 and CD8 in the circulation were significantly higher in thymomas than in the other entities, declined after surgery and in 2 patients the levels increased upon recurrence and then again declined after re-resection. These studies indicate that thymopoiesis, identified by higher IL8 levels in T cells, is present in thymomas. Interestingly the levels in thymic carcinomas were low. The authors conclude that measuring IL8 in T cells could aid in discriminating thymomas that need to be resected, from other mediastinal masses which may not need an operation (e.g. cysts) or require histological diagnosis (germ cell tumors, thymic carcinomas, lymphomas). IL8 levels could not discriminate thymomas from thymic hyperplasia, although imaging may help somewhat.

Although of interest, there are several issues that need to be addressed.

Introduction

- Reference 5 does not report non-therapeutic thymectomy rates.
- Since IL8 is produced by macrophages, some epithelial cells and endothelial cells, it would be important to determine on thymic tissues where IL8 expression is localized.
- Thymic carcinomas belong to the thymic epithelial tumors and although they are more aggressive than thymomas in general, they do represent just the end of the spectrum. B3 thymomas are epithelial cell rich, and they sometimes can hardly be distinguished from thymic carcinomas, in fact an older classification defined them as well differentiated thymic carcinomas. It is interesting and a bit peculiar to note such a big difference in terms of IL8 expression being so low in thymic carcinomas.

Results

- CD31 presence in naïve T cells is definitely less impressive than IL8 and there is a large overlap among subgroups. This may cast some doubt about the IL8 findings.
- Only a few cases were assessed for sjTREC levels in naïve T cells, and not all “negative controls” were included in this analysis. The analysis will need to be expanded.
- The overlap between thymic hyperplasia and thymomas is probably the most concerning finding. With the exception of thymic cysts, all other mediastinal tumors will require either a biopsy or resection.
- The cut-off levels were identified by the ROC analysis. How realistic is the use of these cut-off levels and how could this be implemented in routine workup of mediastinal masses ?
- There were only 2 patients in which a recurrence was associated with increase IL8 expressing naïve T cells. It is hard to make any conclusions based on only these 2 cases. What was the histology of these 2 cases ?
- What do the authors define as “atypical thymoma”?

Discussion

- It is not clear how IL8 level in naïve T cells may help determining whether patients who have MG need a thymectomy.
- Based on only 2 cases of recurrence, suggesting that IL8 levels could be used to monitor thymoma recurrence is premature.

Methods

- The method used to isolate thymocytes from thymic tissues requires more details.

Figures

- In figure 1c there are 2 outliers with extremely high levels of IL8. What histological types were those ? Although overall the median of the levels of IL8 in naïve T cells is significantly higher in thymomas than in the other disorders examined, there is a great level of overlap. Given this, I am concerned that IL8 levels cannot really reliably discriminate these disorders.
- In figure 5, what does it mean “IL8 unchanged”?

NCOMMS-20-03999 “Identification of thymoma with the biomarker cytokine Interleukin-8” by Gao et al.

Summary:

In the revised manuscript, **Fig. 4, Supplementary Fig. 7, Supplementary Fig. 8, Supplementary Fig. 9, Supplementary Fig. 10, Supplementary Fig. 11, Supplementary Table 1 and Supplementary Table 5** (including 3 new cases of thymoma, 2 new case of teratoma, and 1 case of thymic cyst) are new figures/tables that are also shown in this rebuttal letter; and **Fig. 1, Fig. 3, Fig. 4, Fig. 5, Supplementary Fig. 6, Supplementary Table 2, and Supplementary Table 3** are revised figures/tables but are not shown in the letter.

Comments from Reviewer #1

1. Were sample size and power calculations performed for the study? If yes, please describe them in the methods section. If not, please state so in the methods section.

Response:

We thank the Reviewer #1 very much for the valuable suggestions. In order to address this issue, we performed the sample size and power calculations by using R. Based on the results from discovery set, assuming the AUC is 0.95, a sample size of 12 (case: 6, control: 6) is needed to achieve 95% power at 5% significance level. In contrast, in the validation study, with 145 cases in ROC calculation (thymoma cases: 67; non-thymoma cases: 83; healthy controls and thymic hyperplasia are not included into ROC analysis), the power for detecting an AUC of 0.95 is 100%. Therefore, current sample size is enough for the study objective. We will describe sample size and power calculations in the methods section.

2. Please add percentages in the categorical variables presented in Supplementary Table 1. This will enable the reader to compare the composition of the discovery and validation sets. Please check the table for typos. For example the total number of males and females in the validation set exceeds 180.

Response:

We thank reviewer #1 for pointing out these issues. We now add the percentages in **Supplementary Table 1**. And the typos in the **Supplementary Table 1** are corrected (**Reb. Table 1**). It is worth noting that we have added 6 extra cases during the process of revision.

Reb. Table 1. Patient characteristics

Factor	Discovery Set (N=40)	Percentage	Validation Set (N=186)	Percentage
Age, years				
Mean	44.97		47.46	

	Range	20-77		20-77	
Sex					
Male		20	50%	88	47.2%
Female		20	50%	98	52.7%
Healthy Controls		10	25%	33	18.3%
Thymic mass					
Thymoma		10	25%	68	36.6%
Thymic cyst		5	12.5%	48	25.8%
Thymic hyperpla		0	0%	2	1.1%
Thymic		5	12.5%	13	7.0%
Teratoma		5	12.5%	14	7.5%
Lymphoma		5	12.5%	8	4.3%
Thymoma WHO subtype					
A		0	0%	2	2.9%
AB		3	30%	18	26.5%
B1		0	0%	3	4.4%
B2		4	40%	26	38.2%
B3		3	30%	16	23.5%
Thymoma Masaoka stage					
I		1	10%	10	14.7%
II		5	50%	23	33.8%
III		3	30%	25	36.8%
IVa		1	10%	10	14.7%
Myasthenia gravis in thymoma					
Yes		2	20%	26	38.2%
No		8	80%	42	61.8%

3. The manuscript presents a large number of formal statistical comparisons and corresponding p-values without addressing the issue of multiplicity of inferences and controlling the FWER in the study. Please address this issue. For discovery studies, it would be more appropriate to consider a metric such as False Discovery Rate.

Response:

We thank Reviewer #1 for the suggestion. To control False Discovery Rate (FDR) less than 5%, we used Bonferroni adjustment for the multiple comparisons. For example, in the analysis of clinical relevance of IL-8 in the discovery set, the significance level will be 0.0083(0.05/6) based on Bonferroni adjustment. Other statistical comparisons in our study with the issue of multiplicity of inference had also been adjusted .

4. Please provide an estimate of the uncertainty about the cutoff value computed using Yuden's index. Almost none of the IL-8 values in the non-thymoma group reached the threshold in Fig3c.

Response:

We thank Reviewer #1 for the suggestion. To estimate the uncertainty of the cut-off values, we calculated the 95% CI (Confidence interval) for the cut-off values based on non-parametric bootstrap method. The 95% CI for the cut-off values for IL-8⁺ CD4⁺ naïve T cells is 3.55 (2.225, 3.945); The 95% CI for the cut-off values for IL-8⁺ CD8⁺ naïve T cells is 3.35 (1.775, 5).

5. The recommendation presented in Fig 5 includes the use of imaging studies. However, the manuscript does not present information about the combination of imaging with the markers evaluated here.

Response:

Currently, chest CT and MRI with contrast are still the main methods for differential diagnosis of thymic masses, as recommended by NCCN Guideline. IL-8 assay presented in our study would be a supplement to imaging methods. IL-8 cannot be used alone in clinic without imaging manifestation. The combination of imaging with IL-8 assay can significantly improve the sensitivity and specificity of thymic mass diagnosis. According to the NCCN management of thymic masses and IL-8 finding in our study, we therefore put forward such a recommendation in **Fig. 5**.

Comments from Reviewer #2

I have read with interest the manuscript entitled: "Identification of Thymoma with the Biomarker Cytokine Interleukin-8". I think the presented data may support author's conclusions however the manuscript would strongly benefit from far more detailed presentation of the T-cell data as well as an improved (more detailed) methodology section.

Response:

We thank reviewer #2 for positive comments and professional suggestions. We now have provided more detailed T-cell data as well as more detailed methodology to improve our article.

I like the simplicity of the study although find it quite predictable that some thymoma patients display increased production of IL-8 by RTEs T cells despite increased age as compared to age-matched controls (calling it a specific biomarker is another question). Indeed, if well-validated increased levels of RTEs in older patients may be a useful biomarker of specific thymic abnormalities. I speculate, however, that due to extensive variability among different thymoma patients, its usefulness may be limited (especially in some border line cases) and predominantly may apply as a progression marker when

samples from the individual patient could be compared before and after thymic surgery. I stress however that I am not a thoracic clinician.

Response:

We agree that the issue on the variability among different thymoma patients is of great importance. Indeed, thymoma patients with different clinical pathological features of thymoma, including WHO subtypes, tumor sizes, Masaoka stages vary very much in the frequencies of IL-8⁺ T cells. Therefore, the usage of IL-8 has been strictly limited in order to avoid misdiagnosis caused by the variability. However, despite extensive variability, the frequencies of IL-8⁺ T cells in patients with different types of thymomas are significantly higher than those in patients with other thymic tumors IL-8⁺ T cells due to low IL-8 background in healthy adults or patients with other thymic tumors. Therefore, we hope the reviewers agree that IL-8 assay is suitable for thymoma identification.

We agree that, as most diagnostic biomarkers in clinics, IL-8 assay also has a borderline issue. For the diagnosis of these borderline cases, IL-8 assay should be combined with image inspections such as CT and MRI. However, we hope the reviewers agree with us that IL-8 assay alone already achieves very good accuracy rate for thymoma diagnosis. ROC curve analysis shows that IL-8⁺ CD4⁺ T assay alone has a false positive rate of only 7.2% and a false negative rate of only 5.9%, respectively. Given 70/160 (43.7%) misdiagnosis rate by CT diagnosis[1], IL-8 assay is valuable for the auxiliary diagnosis of thymoma.

References Cited:

1. Ackman JB, Verzosa S, Kovach AE, Jr LA, Lanuti M, et al. (2015) High rate of unnecessary thymectomy and its cause. Can computed tomography distinguish thymoma, lymphoma, thymic hyperplasia, and thymic cysts? *European Journal of Radiology* 84: 524-533.

Clinical application of an assay that visualises increased frequency of RTEs as a useful biomarker for thymoma requires in-vitro activation of T cells and intracellular IL-8 staining protocol, which extends to the minimum time of at least 8 hours of processing (is it really a useful in a clinic? How variable the assay may be?).

Response:

As the Reviewer pointed out, IL-8 assay takes ~8 hours of processing, however, IL-8 assay is still practical in clinics. We have discussed with clinicians and laboratory physician to make sure it is available in clinic settings. If patients take blood test in the morning, the test process can be completed within one working day. We agree that repeatability and reproducibility of a biomarker are of great importance for clinical practice. In the IL-8 assay, commercial IL-8 antibodies work very well. To test repeatability and reproducibility, two individuals in our study independently detected IL-8⁺ T cells using the same samples. We found that the variability of IL-8 staining by

different operators is low (**Reb. Fig. 1**). For IL-8⁺CD4⁺ naïve T cells, the Intraclass Correlation Coefficient (ICC) is 0.952, with 95% CI (0.618,0.995). Given that many cytokine assays have been already used for the diagnoses of diseases in clinics, we believe that methodology for IL-8 detection could be further improved for clinical application in the future.

Reb. Fig. 1. Small variability of IL-8 assay. (a) Representative flow cytometry plots of IL-8⁺CD31⁺ cells in CD4⁺ naïve T cells performed by two operators using the same blood samples (left panel); comparison of frequencies of IL-8⁺CD4⁺ naïve T cells in IL-8 assays performed by the two operators (right panel). (b) Representative flow cytometry plots of IL-8⁺CD31⁺ cells in CD8⁺ naïve T cells performed by two operators using the same blood samples (left panel); Comparison of frequencies of IL-8⁺CD8⁺ naïve T cells in IL-8 assays performed by the two operators (right panel).

As authors mentioned there are other markers of recent thymic emigrants and it is essential that those are also incorporated into the study, as they may pose more practical in a clinical setting. Authors discuss the need to test extra surface RTE-markers however they do not address this question (in at least limited number of additional experiments). Authors only test (and compare to IL-8) one surface marker of RTEs: CD31 (PECAM1). CD31 however, has been already demonstrated not to be very useful as a surface marker of bona fide RTEs, based on the fact that CD31⁺ cells already contain a population of peripherally expanded naïve T cells (based on dilution of TCR excision circles (TRECs). Furthermore, CD31⁺ T cells also contain a subpopulation of CD45RA negative memory cells. It is therefore not surprising that surface levels of CD31 do not correlate as well with marking thymomas as does the IL-8 protein expression by RTEs.

Therefore, it would be much more useful to apply direct surface staining (directly on the whole blood and not even PBMCs) using a combination of naïve T cell markers and RTE-markers previously demonstrated to identify cells expressing IL-8 (CXCL8) such as PTK7 as well as complement receptors 2 (CR2). CR2 has been demonstrated

to mark RTEs (also in adults what is especially important when thymoma patients are studied), importantly, when visualized with the bright fluorochrome such as PE and the published CR2 ab clone. CR2 levels will drop after activation and therefore it cannot be used to gate cells secreting IL-8 directly. However, it (IL-8 staining) would not be needed if surface staining biomarker of RTEs would be proven useful (it would streamline the clinical stratification process).

Response:

We thank reviewer #2 for the valuable suggestions. Actually, we have tested some key RTE surface markers including CD31, PTK7, and CR2.

We admit that CD31 alone is not a useful surface marker for thymoma diagnosis as we showed in our manuscript. However, it is still a convincing marker to identify RTEs. To analyze RTE cells, PTK7, IL-8, or CR2 is often combined with CD31[1-3]

PTK7 has been reported to be a classic surface marker of CD4⁺ RTEs. To evaluate the diagnostic performance of PTK7, we made an affinity-purified rabbit anti-human PTK7 IgG antibody as the reference[1] (commercial antibodies do not seem to work well (data not shown)). By using this rabbit antibody, we confirmed that there are more CD31⁺PTK7⁺ naïve CD4⁺ T cells in thymoma patients, compared to patients with other thymic tumors (**Reb. Fig. 2a, b, and Supplementary Fig. 7a, b** in the revised manuscript). However, unfortunately, the ROC curve analysis showed that the area under the ROC curve for the proportion of PTK7⁺ was only 0.85 (**Reb. Fig. 2c, d, and Supplementary Fig. 7c, d** in the revised manuscript), which is lower than that in IL-8 assay

Reb. Fig. 2. The proportions of PTK7⁺CR31⁺ naïve T cells are increased in thymoma patients. The

proportions of PTK7⁺CD31⁺ naïve T cells in PBMCs from patients with thymoma or other thymic masses were analyzed by flow cytometry. **(a)** Representative flow cytometry plots of PTK7⁺CR31⁺ naïve T cells in PBMCs from patients with thymoma and thymic cyst. **(b)** Statistical data of PTK7⁺CD31⁺ naïve T cells in patients with thymoma and other thymic tumors. **(c)** Diagnostic performance of PTK7⁺CD31⁺ naïve T cells in identifying patients with thymomas from patients with other thymic tumors assessed by ROC curve analysis. **(d)** The frequencies of PTK7⁺CD31⁺ naïve T cells in patients with thymomas and other thymic tumors of various ages. The dashed line indicates the optimum cut-off point calculated by applying Youden's J statistic to ROC curves.

We thank the Reviewer #2 very much for the constructive suggestions. CR2 has been recently reported to be a novel marker of RTEs. We also confirmed that CR2 is a RTE marker with good commercial antibodies, and found that there are more CR2⁺CD31⁺CD25⁻ naïve T cells in thymoma patients compared to patients with other thymic tumors (**Reb. Fig. 3 a, b, and Supplementary Fig. 8** in the revised manuscript). However, the area under the ROC curve for the proportion of PTK7⁺ was only 0.84, smaller than that in IL-8 assay.

Reb. Fig. 3. The proportions of CR2⁺ naïve T cells are increased in thymoma patients. The proportions of CR2⁺CD31⁺CD25⁻ naïve T cells from patients with thymoma and other thymic tumors were analyzed by flow cytometry. **(a)** Representative flow cytometry plots of CR2⁺ naïve T cells in PBMCs from patients with thymoma or thymic cyst. **(b)** Statistical data of the proportions of CR2⁺ naïve T cells in patients with thymoma and other thymic tumors. **(c)** Diagnostic performance of CR2⁺ naïve T cells in identifying patients with thymomas from patients with other thymic tumors assessed by ROC curve analysis. **(d)** The frequencies of CR2⁺ naïve T cells in patients with thymomas and other thymic tumors of various ages. The dashed line indicates the optimum cut-off point calculated by applying Youden's J statistic to ROC curves.

Taken together, although IL-8 assay takes a little bit longer time and more work, IL-8

assay has a higher accuracy rate in the diagnosis of thymoma than the assays with reported surface markers such as CD31, PTK7 or CR2.

References Cited:

1. Haines CJ, Giffon TD, Lu LS, Lu X, Tessierlavigne M, et al. (2009) Human CD4⁺ T cell recent thymic emigrants are identified by protein tyrosine kinase 7 and have reduced immune function. *Journal of Experimental Medicine* 206: 275.
2. Das A, Rouault-Pierre K, Kamdar S, Gomez-Tourino I, Wood K, et al. (2017) Adaptive from Innate: Human IFN- γ ⁺ CD4⁺ T Cells Can Arise Directly from CXCL8-Producing Recent Thymic Emigrants in Babies and Adults. *The Journal of Immunology* 199: 1696-1705.
3. Pekalski ML, García AR, Ferreira RC, Rainbow DB, Smyth DJ, et al. (2017) Neonatal and adult recent thymic emigrants produce IL-8 and express complement receptors CR1 and CR2. *Jci Insight*

I find it somehow surprising that followed activation of T cells (6 hours of PMA/ionomycin) authors are still able to use CD31 protein as a gating marker for RTE (part of a gating strategy) as its (PECAM1) levels decrease following activation in-vitro (at least at the RNA level).

Response:

We thank the Reviewer #2 for pointing out these important issues. In our study, we also found that stimulation with PMA plus ionomycin down-regulates the expression level of CD31 on T cells (**Reb. Fig. 4a**). However, the slight down-regulation of CD31 does not significantly affect the gating of CD31⁺ and IL-8⁺ population (**Reb. Fig. 4**) The study published on Nature Medicine[1] (**Fig. 4**) and two other studies[2,3] also took advantage of CD31 protein as a gating marker for RTE in IL-8 assay.

Reb. Fig. 4. PMA/Ionomycin stimulation does not significantly affect the gating of CD31⁺ and IL-8⁺ population. (a) The expression levels of CD31⁺ on naïve T cells in stimulated or unstimulated with PMA plus Ionomycin. (b) The proportion of CD31⁺ within naïve T cells in stimulated sample and unstimulated control group. (c) Co-expression of CD31 and IL-8 in stimulated sample and

unstimulated control group.

References Cited:

1. Gibbons D, Fleming P, Virasami A, Michel M-L, Sebire NJ, et al. (2014) Interleukin-8 (CXCL8) production is a signatory T cell effector function of human newborn infants. *Nature medicine* 20: 1206-1210.
2. Haines CJ, Giffon TD, Lu LS, Lu X, Tessierlavigne M, et al. (2009) Human CD4+ T cell recent thymic emigrants are identified by protein tyrosine kinase 7 and have reduced immune function. *Journal of Experimental Medicine* 206: 275.
3. Pekalski ML, García AR, Ferreira RC, Rainbow DB, Smyth DJ, et al. (2017) Neonatal and adult recent thymic emigrants produce IL-8 and express complement receptors CR1 and CR2. *Jci Insight*.

I wish I could see a more detailed presentation of the raw data, gating strategy of activated and control RTEs: starting from all acquired events through CD3, TCR $\alpha\beta$, CD4, CD8, CD45RA and CD31. This concerns my point above (CD31 down-regulation following activation) as well as for visualisation of possible CD4/CD8 double-positive (DP) population also in the blood of thymoma patients.

Response:

The gating strategy of IL-8⁺ naïve T cells (**Reb. Fig. 6**) has been included in our revised manuscript (**Supplementary Fig. 12**). In our study, we observed few CD4/CD8 double-positive (DP) cells in the blood of thymoma patients (**Reb. Fig. 5**).

Reb. Fig 5. The gating strategy of CD31⁺ naïve T cells

Reb. Fig. 6. The gating strategy of IL-8⁺ naïve T cells. IL-8⁺CD4⁺ naïve T cells were gated as CD235a⁻CD19⁻CD14⁻CD3⁺TCRαβ⁺CD4⁺8⁻CD45RA⁺CCR7⁺CD31⁺IL-8⁺. IL-8⁺CD8⁺ naïve T cells were gated as CD235a⁻CD19⁻CD14⁻CD3⁺TCRαβ⁺CD4⁺8⁺CD45RA⁺CCR7⁺CD31⁺IL-8⁺. CD14 was taken into gating strategy to exclude the inference of TCR⁺ macrophages.

There is an extensive variability within the frequency of IL-8 positive T cells among the group of thymoma patients- I wish authors could better describe what could potentially explain this variability, as currently they only discuss this matter without detailed data presentation of IL-8⁺ T cells frequency vs different variables studied (tumour mass etc). Paper will benefit from a more advanced description of thymoma stages and perhaps histopathological data.

Response:

We thank the Reviewer #2 for the valuable comments and suggestions. To address the variability issue, we re-analyzed our data in more detail as shown in **Reb. Fig. 7** (also see **Supplementary Fig. 9** in the revised manuscript). We found that clinical pathological features of thymoma, including WHO subtypes, tumor sizes, Masaoka stages potentially contribute to the variability within the frequency of IL-8⁺ T cells among the group of thymoma patients.

The frequencies of IL-8 positive T cells in patients with “lymphocyte-rich” type B2 thymomas are obviously higher than those in patients with “lymphocyte-poor” type A and B3, confirming that IL-8 positive RTE cells are associated with the potency of intratumorous thymopoiesis (**Reb. Fig. 7a**). We also observed higher frequencies of IL-

8⁺ T cells in patients with Masaoka stage III and IVa thymoma, compared to patients with Masaoka stage I and II thymoma (**Reb. Fig. 7c-d**). In addition, our data show positive correlation between the frequencies of IL-8⁺ T cells and the tumor sizes of thymoma. Patients with larger thymoma have higher frequency of IL-8⁺ T cells (**Reb. Fig. 7e-f**). In our study, tumor sizes of thymoma range from 1.5 cm diameter to 13 cm diameter. Such a big variability may also contribute to extensive variability within the frequency of IL-8 positive T cells.

We have included these data in our revised manuscript and made a more advanced description of the relationship between clinical pathological features and IL-8 in revised manuscript.

Reb. Fig. 7. The variability of IL-8⁺ naive T cells within thymoma patients is caused by clinical pathological features. (a) The relationship between IL-8⁺CD4⁺ naive T cells and WHO subtypes. **(b)** The relationship between IL-8⁺CD8⁺ naive T cells and WHO subtypes. **(c)** The relationship between IL-8⁺CD4⁺ naive T cells and Masaoka stages. **(d)** The relationship between IL-8⁺CD8⁺ naive T cells and Masaoka stages. **(e)** The relationship between IL-8⁺CD4⁺ naive T cells and tumor sizes. **(f)** The relationship between IL-8⁺CD8⁺ naive T cells and tumor sizes. The summary data in were presented as mean ± SD Statistical differences were determined by One-way ANOVA with Bonferroni adjustment and q values were indicated by * (p<0.05), or ** (p<0.01), or *** (p<0.001).

Overall I think the manuscript contains interesting scientific and clinical information

and has a potential for publication when all questions are carefully addressed.

Response:

We thank the reviewer for encouraging comments.

Comments from Reviewer #3

Although of interest, there are several issues that need to be addressed.

Response:

We thank Reviewer #3 for acknowledging our study being interesting, and we have carefully addressed the issues that the editors and reviewers pointed out.

Reference 5 does not report non-therapeutic thymectomy rates.

Response:

We apologize for our mistake, and we now have corrected it.

Since IL8 is produced by macrophages, some epithelial cells and endothelial cells, it would be important to determine on thymic tissues where IL8 expression is localized.

Response:

We agree with the Reviewer #3 that this is an important issue. We have taken it into consideration that other lineage cells expressing IL-8 such as monocytes in peripheral blood may interfere with IL-8 assay. Therefore, we included key markers of these lineages such as CD14 (monocytes and macrophages), CD235a (erythroid precursors and erythrocytes), and CD19 (B cells) in our staining to exclude these cells (**Reb. Fig. 6**).

Given that IL-8-producing CD4⁺ T cells are enriched among RTEs, it is reasonable to ask where IL-8 expression is localized in thymic tissues. However, our study and a previous study by others [1] show that thymocytes do not express or express very low level of IL-8 compared with T cells in periphery. (**Reb. Fig. 8**).

Reb. Fig. 8. The expression of IL-8 in thymocytes in thymoma and T cells in blood. The expression levels of IL-8 in CD4⁺CD8⁺ DP thymocytes, CD4⁺CD8⁺ SP thymocytes, and CD4⁺CD8⁻ SP thymocytes in thymoma, and naïve CD4⁺T cells and naïve CD8⁺T cells in PBMCs from the same thymoma patient were examined by Flow cytometry after PMA and Ionomycin stimulation.

References Cited:

1. Das A, Rouault-Pierre K, Kamdar S, Gomez-Tourino I, Wood K, et al. (2017) Adaptive from Innate: Human IFN- γ + CD4⁺ T Cells Can Arise Directly from CXCL8-Producing Recent Thymic Emigrants in Babies and Adults. *The Journal of Immunology* 199: 1696-1705.

Thymic carcinomas belong to the thymic epithelial tumors and although they are more aggressive than thymomas in general, they do represent just the end of the spectrum. B3 thymomas are epithelial cell rich, and they sometimes can hardly be distinguished from thymic carcinomas, in fact an older classification defined them as well differentiated thymic carcinomas. It is interesting and a bit peculiar to note such a big difference in terms of IL8 expression being so low in thymic carcinomas.

Response:

The issue raised by this reviewer is quite important. In our study pathological diagnosis of all the cases in our study was made by qualified pathologists in our hospital.

Type B3 thymoma is an epithelium-predominant thymic tumor composed of mildly or moderately atypical polygonal tumor cells showing a sheet-like, solid growth pattern. There are intermingled immature T cells in most of cases. IHC phenotypes: The tumor cells usually react with pan-cytokeratin, and express CK19, CK5/6, CK7, CK8, and CK19, but not CK20, p63, and PAX8. Thymic carcinoma markers are almost always negative (CD5, CD117), or focally expressed in rare cases (GLUT1 and MUC1). Terminal deoxynucleotidyl transferase (TdT)+ immature T cells occur in >95% of B3 thymoma cases.

Thymic squamous cell carcinoma is a malignant neoplasm of thymus with morphological features of squamous cell carcinoma as seen in other organs. It consists of infiltrative sheets, islands, and cords of large polygonal cells, accompanied by broad zones of desmoplastic to sclerohyaline stroma that is variably infiltrated by chronic inflammatory cells. Unlike thymomas, it generally lacks resemblance to normal thymic cytoarchitecture, such as discrete lobulation, perivascular spaces, and admixed immature T lymphocytes. IHC phenotypes: CD5, CD117, GLUT1 and MUC1 are frequently expressed in thymic carcinoma (approximately 80%), but much less common in thymomas, and therefore may be of value in differential diagnosis of difficult cases.

According to these pathological features, although B3 thymomas are “epithelial cell rich” and “lymphocyte-poor”, they still bear a few immature T cells. Tumor cells in

thymic carcinoma totally lost the function of supporting T cell development, which has been proved by the rare occurrence of TdT⁺ T cells. Therefore, it is not surprising that thymic carcinomas seldomly export newly developed T cells, and exhibit lower level of IL-8 compared to B3 thymomas. We have included more detailed data about IL-8⁺ T cells in different WHO subtypes in our revised manuscript (**Reb Fig. 9**).

Reb. Fig. 9. The frequency of IL-8⁺ naïve T cells in thymoma patients with different WHO subtypes and thymic carcinoma patients. (a) The statistical data of the frequencies of IL-8⁺CD4⁺ naïve T cells in thymoma and thymic carcinoma patients. (b) The statistical data of the frequencies of IL-8⁺CD8⁺ naïve T cells in thymoma and thymic carcinoma patients. The summary data in were presented as mean ± SD. Statistical differences were determined by One-way ANOVA with Bonferroni adjustment and q values were indicated by * (p<0.05), or ** (p<0.01), or *** (p<0.001).

CD31 presence in naïve T cells is definitely less impressive than IL8 and there is a large overlap among subgroups. This may cast some doubt about the IL8 findings.

Response:

CD31 is not a specific marker of RTE cells. RTE cells are enriched in the CD31⁺ cell population, that is to say, not all the CD31⁺ cells are RTE cells. In contrast, IL-8 is a specific marker of RTE cells. Therefore, it is not surprising that CD31 presence in naïve T cells is less impressive than IL8, as mentioned by the Reviewer #2.

Only a few cases were assessed for sjTREC levels in naïve T cells, and not all “negative

controls” were included in this analysis. The analysis will need to be expanded.

Response:

We thank the Reviewer for the previous suggestion. We have expanded the analysis of sjTrecs levels in patients with different thymic masses, including thymoma, thymic cyst, teratoma and carcinoma. We are now seeking more cases with lymphoma. The work will be finished as soon as possible.

The overlap between thymic hyperplasia and thymomas is probably the most concerning finding. With the exception of thymic cysts, all other mediastinal tumors will require either a biopsy or resection.

Response:

Since IL-8 level is elevated in both thymic hyperplasia and thymoma, IL-8 assay alone cannot distinguish thymic hyperplasia from thymoma. However, thymic hyperplasia has some typical imaging manifestations on CT scan or MRI. Clinically, we use chest CT/MRI with contrast to identify thymic hyperplasia from thymoma.

The cut-off levels were identified by the ROC analysis. How realistic is the use of these cut-off levels and how could this be implemented in routine workup of mediastinal masses ?

Response:

Clinically, we mainly use IL-8 assay to assist in the diagnosis of thymoma, thymic cyst, and lymphoma. For patients with thymoma or cyst, patients whose IL-8 levels are higher than 95% CI (Confidence interval) of the cut-off value very likely have thymoma and can receive a resection; patients whose IL-8 level is lower than 95% CI of the cut-off value are diagnosed with thymic cyst and receive follow-ups. For patients with thymoma or lymphoma, patients whose IL-8 levels are higher than 95% CI of the cut-off value are diagnosed with thymoma and receive a resection; patients whose IL-8 levels are lower than 95% CI (Confidence interval) are probably diagnosed with lymphoma and receive a biopsy.

There were only 2 patients in which a recurrence was associated with increase IL8 expressing naïve T cells. It is hard to make any conclusions based on only these 2 cases. What was the histology of these 2 cases ?

Response:

We agree that two cases of thymoma recurrence are too few to make solid conclusions. In our revised manuscript, we included two more patients who were recently diagnosed with thymoma recurrence. The histology of all four cases can be seen in **Reb. Table 2**. Consistent with our previous results, the two patients also showed increased frequencies of IL-8⁺ naïve T cells when tumor recurred (**Reb. Fig. 10**, also see **Fig. 4 and**

Supplementary Fig. 9 in the revised manuscript), confirming that IL-8 assay can be applied to the active surveillance of thymoma recurrence. We will continue to keep in touch with these two patients to evaluate their IL-8 level after second surgery. Thymoma is not a very common malignant tumor, with only an incidence of 1.5/1000000. Moreover, the recurrence rate of thymoma is quite low. Therefore, we hope that four cases of thymoma recurrence would be acceptable.

Reb. Fig. 10. Thymoma patients have increased frequencies of IL-8⁺ naïve T cells when thymoma recurred. (a) The representative flow cytometry plots (left panel) and statistical data (right panel) of IL-8⁺CD31⁺ cells in CD4⁺ naïve T cells or CD8⁺ naïve T cells in PBMCs at different time points before and after the first thymoma resection in Case #3 patient with thymoma recurrence. (b) The representative flow cytometry plots (left panel) and statistical data (right panel) of IL-8⁺CD31⁺ cells in CD4⁺ naïve T cells or CD8⁺ naïve T cells in PBMCs at different time points before and after the first thymoma resection in Case #4 patient with thymoma recurrence.

Reb. Table 2. Clinical characteristics of the four cases with thymoma recurrence

Case No.	Gender	Age	Pathological diagnosis	WHO subtype	Tumor size

1	Female	55	Thymoma	B3	7
2	Male	51	Thymoma	B2	9
3	Male	56	Thymoma	B2	2.5
4	Female	62	Thymoma	B2	4

What do the authors define as “atypical thymoma” ?

Response:

We thank the Reviewer #3 for kindly pointing out this issue. In our manuscript, “atypical thymoma” referred to thymoma that does not have the typical imaging findings of thymoma. We already changed it to “not typical thymoma” in the revised manuscript.

It is not clear how IL8 level in naïve T cells may help determining whether patients who have MG need a thymectomy.

Response:

This is a constructive suggestion. In this study, we found IL-8 can reflect the thymopoiesis state. Although the cause of MG is still unclear, it is related to the thymopoiesis state because thymectomy can improve the clinical outcomes of patients with non-thymomatous MG [1]. Therefore, we speculate that MG patients with elevated IL-8 level may have enhanced thymopoiesis, and it may be more likely that these patients could benefit from thymectomy. Further study is needed to address this issue.

References Cited:

1. Wolfe GI, Kaminski HJ, Aban I, Minisman G, Kuo H, et al. (2016) Randomized Trial of Thymectomy in Myasthenia Gravis. The New England Journal of Medicine 375: 511-522.

Based on only 2 cases of recurrence, suggesting that IL8 levels could be used to monitor thymoma recurrence is premature.

Response:

We agree that two cases of thymoma recurrence are too few to make solid conclusions. We now included two more cases with thymoma recurrence (**Reb. Fig. 10**, also see **Fig. 4 and Supplementary Fig. 9** in the revised manuscript).

The method used to isolate thymocytes from thymic tissues requires more details.

Response:

We have added a detailed description of thymocytes isolation in the revised manuscript (please see **Isolation of PBMCs and thymocytes** in **Methods**).

In figure 1c there are 2 outliers with extremely high levels of IL8. What histological types were those ?

Response:

We thank the Reviewer #3 for this important point. We checked the information of these two patients. The patient characteristics were listed in the **Reb. Table 3**. The detailed information was also included in our revised manuscript (**Supplementary Table 5**). The two cases of thymomas are both type B2 thymoma. Both of their tumor sizes are extremely large. The two cases confirm that IL-8 level is very positively correlated with tumor size. Since B2 thymoma is lymphocyte-rich thymomas, it is not strange that cases with such a large B2 thymoma have extremely high frequencies of IL-8⁺ T cells.

Reb. Table 3. Clinical characteristics of the two patients with extremely high levels of IL8

Case No.	Gender	Age	Pathological diagnosis	WHO subtype	Tumor size	IL-8 in CD4 ⁺ naïve	IL-8 in CD8 ⁺ naïve
1	Male	47	Thymoma	B2	10	30.6	27.2
2	Male	57	Thymoma	B2	11.3	29.1	24.4

Although overall the median of the levels of IL8 in naïve T cells is significantly higher in thymomas than in the other disorders examined, there is a great level of overlap. Given this, I am concerned that IL8 levels cannot really reliably discriminate these disorders.

Response:

We admit that there are a few overlapping cases between thymomas and other thymic tumors. We hope the Reviewer agree with us that IL-8 assay alone already achieves very good accuracy rate for thymoma diagnosis. ROC curve analysis shows that IL-8⁺ CD4⁺ T assay alone has a false positive rate of only 7.2% and a false negative rate of only 5.9%, respectively. Given 70/160 (43.7%) misdiagnosis rate by CT diagnosis[1], IL-8 assay can greatly improve accuracy of diagnosis of thymic tumors.

The “borderline” issue is not rare for diagnostic markers in clinics. For these cases, they usually are diagnosed by combination of different inspections and clinical symptoms. For thymoma diagnosis of borderline cases, IL-8 assay should be combined with image inspections such as CT and MRI. Although the accuracy of combining IL-8 to identify thymomas can not reach 100%, it is still much higher than that of chest CTs/MRIs alone. We hope the reviewer could agree with us that our study is practical for clinical utility.

In figure 5, what does it mean “IL8 unchanged” ?

Response:

We apologize for the misleading description. “IL-8 unchanged” means that the

frequency of IL-8⁺ T cells is similar to that in healthy patients. We have revised it in our revised manuscript (**Fig. 5** in the revised manuscript).

REVIEWER COMMENTS

Reviewer #1 (Remarks to the Author):

1. Thank you for the detailed responses to my comments on the previous version of the manuscript.
2. Based on the response to my question about sample size and power computations, it seems that these were developed in time for this revision of the manuscript and were not part of the planning of the study. Such post-hoc computations are not of importance to the investigation. I recommend to delete the sentence on sample size computations that was added to the statistical analysis section. If you retain the calculations in the appendix, please label them clearly as "post-hoc".
3. Please note that the Bonferroni adjustment is used to control the FWER in the study and not FDR. If you intend to control FWER that way, I assume you have considered the fact that the Bonferroni adjustment is known to be rather conservative. Please specify which formal comparisons are covered by the Bonferroni adjustment. For all other unadjusted comparisons, it would be appropriate to not report p-values but only unadjusted confidence intervals. You should indicate that these intervals have not been adjusted for multiplicity.
4. Please provide a reference to a publication describing the method used to calculate the confidence interval for the cutoff value.
5. Insofar as the manuscript does not present actual data comparing imaging alone to the combination of imaging and the markers evaluated here, the recommendation shown in Fig 5 does not seem to be evidence-based. If you retain it, I suggest to present your proposal as , perhaps, considered opinion.

Reviewer #2 (Remarks to the Author):

Dear Authors,

Thank you for your responses to my queries and for the additional work that you have done to validate other RTE-markers in thymoma.

I feel your answers fulfil my question and I am happy for the manuscript to be taken forward.

I suggest changing the sentence in the discussion to make it more obvious that IL-8/other RTE markers are of limited potential as thymoma markers in not only children or adolescents (as already stated within the discussion) but also in adults with remaining thymic tissue (up to 40 y.o.). Currently, the discussion somehow ignores the utilisation of your CXCL8-assay as thymoma marker within the age gap between adolescent and 40y.o.

Perhaps the title should also state that your findings are of the highest importance for thymoma patients who are older than 40y.o. to avoid misdiagnosis.

Best wishes,

Marcin Pekalski

Reviewer #3 (Remarks to the Author):

The authors have adequately addressed my concerns.

Summary:

The newly revised paper includes 4 figures, 2 tables, 11 supplementary figures, and 3 supplementary tables (in contrast to the previous version that has 5 figures and 11 supplementary figures, and 5 supplementary tables). The original **Fig. 5** was moved into the new supplementary file as **Supplementary Fig. 10**, and the **Supplementary Table 1** and the **Supplementary Table 2** in the previous version are **Table 1** and **Table 2** in the revised main text, respectively. **Supplementary Fig. 2** and **Supplementary Fig. 4** are revised figures (we expanded sjTREC analysis, as Reviewer #3 suggested in the previous comments).

Comments from Reviewer #1

1. Thank you for the detailed responses to my comments on the previous version of the manuscript.

Based on the response to my question about sample size and power computations, it seems that these were developed in time for this revision of the manuscript and were not part of the planning of the study. Such post-hoc computations are not of importance to the investigation. I recommend to delete the sentence on sample size computations that was added to the statistical analysis section. If you retain the calculations in the appendix, please label them clearly as “post-hoc” .

Response:

We thank Reviewer #1 again for the professional statistical suggestions. Our manuscript has benefited much from these kind suggestions.

In the revised manuscript, we have deleted the sentence “Sample size and power calculations were performed by using R.” in the “Statistical analysis” section. Please see Page 12 in the revised manuscript.

2. Please note that the Bonferroni adjustment is used to control the FWER in the study and not FDR. If you intend to control FWER that way, I assume you have considered

the fact that the Bonferroni adjustment is known to be rather conservative. Please specify which formal comparisons are covered by the Bonferroni adjustment. For all other unadjusted comparisons, it would be appropriate to not report p-values but only unadjusted confidence intervals. You should indicate that these intervals have not been adjusted for multiplicity.

Response:

We thank Reviewer #1 for pointing out the important issue. We agree with Reviewer #1 that the Bonferroni adjustment is to control FWER and not FDR, and it would be more appropriate to control FDR for discovery studies. In the revised manuscript, we took advantage of the Benjamini-Hochberg method to control FDR. As shown in the **Reb. Table 1**, all P values that were less than 0.05 before adjustments are still below 0.05 after adjustments with the Benjamini–Hochberg procedure.

Reb. Table 1. Adjusted P values by the Benjamini–Hochberg procedure

NO.	Tables/ Figures	Description	Unadjusted P value	Adjusted P value*
1	Figure 1a	Increased IL-8 ⁺ CD4 ⁺ T% in thymoma group	2.00E-04	7.57E-04
2	Figure 1b	Increased IL-8 ⁺ CD8 ⁺ T% in thymoma group	4.00E-04	0.0013
3	Figure 1c	Increased IL-8 ⁺ CD4 ⁺ T% in thymoma group	3.82E-24	1.45E-22
4	Figure 1d	Increased IL-8 ⁺ CD8 ⁺ T% in thymoma group	6.32E-21	1.20E-19
5	Figure 2a	Decreased IL-8 ⁺ CD4 ⁺ T% after surgery in thymoma group	2.48E-07	3.14E-06
6	Figure 2a	No change in IL-8 ⁺ CD4 ⁺ T% after surgery in cyst group	0.1840	0.2590
7	Figure 2a	No change in IL-8 ⁺ CD4 ⁺ T% after surgery in carcinoma group	0.5001	0.5639
8	Figure 2a	No change in IL-8 ⁺ CD4 ⁺ T% after surgery in teratoma group	0.5930	0.6259
9	Figure 2a	No change in IL-8 ⁺ CD4 ⁺ T% after surgery in lymphoma group	0.999	0.9990
10	Figure 2b	Decreased IL-8 ⁺ CD8 ⁺ T % after surgery in thymoma group	3.82E-07	3.63E-06
11	Figure 2b	No change in IL-8 ⁺ CD8 ⁺ T% after surgery in cyst group	0.2003	0.2718
12	Figure 2b	No change in IL-8 ⁺ CD8 ⁺ T% after surgery in carcinoma group	0.3452	0.4271
13	Figure 2b	No change in IL-8 ⁺ CD8 ⁺ T% after surgery in teratoma group	0.1088	0.1654
14	Figure 2b	No change in IL-8 ⁺ CD8 ⁺ T% after surgery in lymphoma group	0.2850	0.3734
15	Table 2	No difference in IL-8 ⁺ CD4 ⁺ T% between male and female	0.1378	0.2014
16	Table 2	Higher IL-8 ⁺ CD4 ⁺ T% in WHO B2 and B3 thymoma patients	0.0269	0.0426
17	Table 2	No difference in IL-8 ⁺ CD4 ⁺ T% between thymomas with or without myasthenia gravis	0.5045	0.5639
18	Table 2	Higher IL-8 ⁺ CD4 ⁺ T% in thymomas at Masaoka stages III and IVa	0.0138	0.0248

19	Table 2	No difference in IL-8 ⁺ CD4 ⁺ T% in thymoma patients of different age ranges	0.3484	0.4271
20	Table 2	Higher IL-8 ⁺ CD4 ⁺ T% in patients with larger thymomas	9.36E-04	0.0024
21	Table 2	No difference in IL-8 ⁺ CD8 ⁺ T% between male and female	0.3851	0.4573
22	Table 2	Higher IL-8 ⁺ CD8 ⁺ T% in WHO B2 and B3 thymoma patients	0.0147	0.0248
23	Table 2	No difference in IL-8 ⁺ CD8 ⁺ T % between thymomas with or without myasthenia gravis	0.9307	0.9559
24	Table 2	Higher IL-8 ⁺ CD8 ⁺ T% in thymomas at Masaoka stages III and IVa	0.0126	0.0248
25	Table 2	No difference in IL-8 ⁺ CD8 ⁺ T% in thymoma patients of different age ranges	0.5345	0.5803
26	Table 2	Higher IL-8 ⁺ CD8 ⁺ T% in patients with larger thymomas	1.53E-05	9.69E-05
27	Figure S2b	Increased CD31 ⁺ CD4 ⁺ T% in thymoma patients compared to patients with other thymic tumors in the discovery set	0.0150	0.0248
28	Figure S2c	Higher sjTRECs levels in naïve CD4 ⁺ T in thymoma group	1.44E-05	9.69E-05
29	Figure S2c	Higher sjTRECs levels in naïve CD8 ⁺ T in thymoma group	2.19E-04	7.57E-04
30	Figure S4a	Increased CD31 ⁺ CD4 ⁺ T% in thymoma group in the validation set	2.03E-05	1.10E-04
31	Figure S6a	Increased PTK7 ⁺ CD4 ⁺ T% in thymoma group	4.22E-05	2.00E-04
32	Figure S7a	Increased CR2 ⁺ CD4 ⁺ T% in thymoma group	5.00E-04	0.0015
33	Figure S9a	Higher IL-8 ⁺ CD4 ⁺ T% in B2 thymoma patients compared to B3 thymoma patients	9.00E-04	0.0024
34	Figure S9a	Higher IL-8 ⁺ CD4 ⁺ T% in B2 thymoma patients compared to A,B and B1 thymoma patients	0.0015	0.0036
35	Figure S9b	Higher IL-8 ⁺ CD8 ⁺ T% in B2 thymoma patients compared to B3 thymoma patients	0.0018	0.0040
36	Figure S9b	Higher IL-8 ⁺ CD8 ⁺ T% in B2 thymoma patients compared to A,B and B1 thymoma patients	2.00E-04	7.57E-04
37	Figure S9c	Higher IL-8 ⁺ CD4 ⁺ T% in patients at stage III and IVa compared to patients at stage I or II	0.0138	0.0248
38	Figure S9d	Higher IL-8 ⁺ CD8 ⁺ T% in patients at stage III and IVa compared to patients at stage I or II	0.0126	0.0248

* P values were adjusted with the Benjamini–Hochberg procedure.

3. Please provide a reference to a publication describing the method used to calculate the confidence interval for the cutoff value.

Response:

We have cited two references (Reference 36 and 37, respectively) in the newly revised manuscript as follows:

- 1) Robin X, Turck N, Hainard A, Tiberti N, Lisacek F, Sanchez J-C, et al.

pROC: an open-source package for R and S+ to analyze and compare ROC curves. *BMC Bioinformatics*. 2011, 12(1): 1-8.

2) Carpenter J, Bithell J. Bootstrap confidence intervals: when, which, what? A practical guide for medical statisticians. *Stat Med*. 2000, 19(9): 1141-1164.

4. Insofar as the manuscript does not present actual data comparing imaging alone to the combination of imaging and the markers evaluated here, the recommendation shown in Fig 5 does not seem to be evidence-based. If you retain it, I suggest to present your proposal as, perhaps, considered opinion.

Response:

We agree with Reviewer #1 that the recommendation shown in the Fig 5 in previous version is not evidence-based. We have moved “Recommendation of IL-8 Evaluation for Differential Diagnosis of Thymic Masses” from the Result section to the Discussion section. Please see Page 9 in the newly revised manuscript.

Comments from Reviewer #2

Dear Authors,

Thank you for your responses to my queries and for the additional work that you have done to validate other RTE-markers in thymoma. I feel your answers fulfil my question and I am happy for the manuscript to be taken forward.

I suggest changing the sentence in the discussion to make it more obvious that IL-8/other RTE markers are of limited potential as thymoma markers in not only children or adolescents (as already stated within the discussion) but also in adults with remaining thymic tissue (up to 40 y.o.). Currently, the discussion somehow ignores the utilisation of your CXCL8-assay as thymoma marker within the age gap between adolescent and 40y.o.

Perhaps the title should also state that your findings are of the highest importance for thymoma patients who are older than 40y.o. to avoid misdiagnosis.

Best wishes,

Marcin Pekalski

Response:

We thank Dr. Marcin Pekalski for his encouraging remarks and pointing out the important issue. We agree with Dr. Marcin Pekalski that the age gap between adolescent and 40 years old in the utilisation of IL8-assay should not be ignored. In our revised manuscript, we discussed more about this issue pointed out by Dr. Marcin Pekalski and made it more obvious in the Discussion that IL-8/other RTE markers are of limited potential as thymoma markers in not only children or adolescents but also in adults with remaining thymic tissue. Please see Page 9 in the newly revised manuscript.

Comments from Reviewer #3

The authors have adequately addressed my concerns.

Response:

We thank the Review #3 very much for the favorable comment.

REVIEWERS' COMMENTS:

Reviewer #1 (Remarks to the Author):

Thank you for the responses to my previous review and corresponding edits to the manuscript. I have no further comments.